# Quantitative Conspicuity of Pancreatic Canine Insulinoma: A Comparison of Dynamic 4D CT and Dual-Source, Dual-Energy Bolus-Triggered Multiphase CT Imaging

**DOI:** 10.3390/vetsci12111102

**Published:** 2025-11-19

**Authors:** Veronica Camosci, Claudia Canton, Laura Ventura, Giovanna Bertolini

**Affiliations:** 1San Marco Veterinary Clinic and Laboratory, Diagnostic and Interventional Radiology Division, 35030 Veggiano, Italy; veronica.camosci@sanmarcovet.it (V.C.); claudia.canton@sanamarcovet.it (C.C.); 2Department of Statistical Sciences, University of Padova, 35121 Padova, Italy; ventura@stat.unipd.it

**Keywords:** canine insulinoma, insulinoma detection, perfusion CT, DECT, conspicuity

## Abstract

Insulinoma is a rare pancreatic endocrine tumor that causes hypoglycemia and represents the most common functional pancreatic neuroendocrine tumor in both humans and dogs. Accurate preoperative staging using diagnostic imaging, particularly contrast-enhanced computed tomography (CT), is crucial for prognosis and surgical planning. In dogs, as in humans, insulinomas are typically small with variable enhancement patterns, sometimes showing transient hyperenhancement, which complicates detection. Advances in CT technology, including perfusion CT and dual-energy CT (DECT), have improved lesion characterization. Perfusion CT provides time-resolved perfusion data, useful for identifying tumors with unpredictable enhancement. However, studies on canine insulinomas remain limited. This retrospective study compares the conspicuity and CT characteristics of pancreatic insulinomas in dogs using multiphase dual-energy CT (DECT) and dynamic 4D perfusion CT, aiming to assess their relative value for accurate detection and characterization to support preoperative staging and surgical planning.

## 1. Introduction

Insulinoma is a rare neuroendocrine tumor that causes hypersecretion of insulin, leading to hypoglycemia [1] and represents the most frequent canine pancreatic neuroendocrine tumor, with an annual prevalence of 0.004% in dogs under primary veterinary care in the UK in 2019 [2]. In veterinary medicine contrast-enhanced computed tomography (CECT) is the preferred method for staging canine insulinoma and is essential for planning surgical excision, the most effective treatment [3]. Preoperative staging of insulinomas using diagnostic imaging techniques is of fundamental importance. First, it provides valuable prognostic information since dogs with insulinomas with distant metastases have shorter survival times. Secondly, it can also provide detailed information about the location of the insulinoma within the pancreas, which is an important part of surgical planning [4].

In human medicine, computed tomography (CT) is routinely used to localize insulinomas and assess metastatic spread [5]. These tumors are typically small, often causing only subtle or no changes in pancreatic contour, so their detection largely relies on enhancement patterns [1]. Such patterns are variable and time-dependent, with contrast between tumor and surrounding parenchyma fluctuating over time [6]. Fu et al. [7] reported in 2020 that the timing of optimal tumor visibility varies among individuals, with 30.2% of insulinomas displaying a rapid, transient increase in contrast during the arterial phase. More recently, the tumor-to-pancreas ratio (TPR) and contrast-to-noise ratio (CNR) have been proposed as quantitative measures to assess optimal conspicuity—i.e., how well a lesion is visualized relative to the surrounding parenchyma—in pancreatic carcinomas [8]. Advances in CT technology, including perfusion CT and dual-energy CT (DECT), may further improve pancreatic imaging [9].

Perfusion CT is a relatively recent technique, which can provide qualitative and quantitative information on tissue perfusion parameters in a non-invasive manner [10]. Dynamic volumetric perfusion CT (dynamic 4D CT) acquires datasets at multiple closely spaced time points, generating temporal attenuation curves (TACs) for each voxel. This approach allows determination of the optimal time window for imaging acquisition of lesions with unpredictable enhancement, such as insulinomas showing rapid early arterial-phase contrast uptake [11]. In dogs, pancreatic perfusion CT has been studied in acute pancreatitis, showing a shorter time-to-peak enhancement compared to healthy controls [12], but its application in pancreatic neoplasms has not yet been explored.

DECT offers several advantages for pancreatic imaging, including virtual monoenergetic imaging (VMI) and virtual non-contrast imaging (VNC). VMI uses dual-energy data to create images at multiple simulated energy levels, enhancing iodine-containing tissues at lower kiloelectronvolts due to increased photoelectric absorption, while higher kiloelectronvolts reduce image noise, balancing image quality and radiation exposure [13,14]. VNC exploits differences in X-ray absorption to separate or remove materials such as calcium or iodine, potentially replacing a true non-contrast scan while maintaining adequate diagnostic quality [13].

The available literature on the CT appearance of insulinomas in dogs includes studies with a limited number of patients and a wide range of different machines and acquisition protocols, leading to disparate results mostly describing iso/hypoattenuating appearance in pre-contrast series and hyperenhancement during the arterial phase [4,15,16,17,18].

This study aims (1) to describe dynamic 4D CT and DECT in a large cohort of canine patients with suspected insulinoma and compare the diagnostic performance of dynamic 4D CT with DECT at different energy levels in lesion detection and localization via conspicuity calculation of tumor-to-pancreas ratio (TPR) and contrast-to-noise ratio (CNR) [7]; (2) report morphological, invasive and metastatic CT characteristics of canine insulinomas.

## 2. Materials and Methods

### 2.1. Patient Selection

This is a single-center retrospective study, conducted during a 7-year period, from 2016 to 2023. All procedures were performed at the Division of Diagnostic and Interventional Radiology of the San Marco Veterinary Clinic exclusively for the benefit of the patient and for standard diagnostic and monitoring purposes. Prior written informed consent was obtained from all dog owners. All procedures performed were in accordance with European legislation “on the protection of animals used for scientific purposes” (Directive 2010/63/EU) and the ethical requirements of Italian law (Legislative Decree 4 March 2014, n. 26). The database of the San Marco Veterinary Clinic and Laboratory was retrospectively searched for patients diagnosed with insulinoma. Inclusion criteria for eligible patients were set as follows: (1) a cytological or histopathological diagnosis of insulinoma or, in its absence, clear clinical and imaging findings consistent with pancreatic insulinoma. All dogs included here were assessed by specialists of the Internal Medicine Department of San Marco Veterinary Clinic and Laboratory and underwent a complete clinical evaluation, including comprehensive clinicopathological, endocrinological, and imaging assessments. Insulinoma was suspected in cases of persistent hypoglycemia without other identifiable causes, together with concurrently inappropriate serum insulin levels. Blood glucose and insulin values are provided in the Appendix A, Appendix A. (2) A reviewable whole-body CT acquired using a 192x2 dual-source, dual-energy CT scanner (Somatom Force 192x2; Siemens, Erlangen, Germany)). (3) Scan protocol including a Dynamic 4D CT or a bolus-triggered, multiphase dual-energy CT series. Exclusion criteria included incomplete CT data or acquisition via other scan protocols. CT examinations of patients selected were retrieved from the PACS (Picture Archiving Communication System) and analyzed using dedicate freestanding workstation and vendor-specific postprocessing software (Syngo.Via, VB60S_HF01 Siemens, Germany).

### 2.2. CT Technique

All dogs were positioned in sternal recumbency, headfirst and forelimbs extended cranially. The iodinated contrast agent (Visipaque 320 mg/mL, dosage of 2 mL/kg) was injected into a cephalic vein each time at an injection rate of 2 mL/s using a double-barrel injection system followed by a saline flush at the same contrast volume and injection rate of the contrast (dosage of 2 mL/kg and an injection rate of 2 mL/s). The anesthesia protocol was tailored according to the patient’s health condition and comorbidities. Sedation included intramuscular methadone (0.2 mg/kg); in some cases, a low dose of dexmedetomidine (0.5–2 mcg/kg) was administered. Anesthesia was induced with intravenous propofol and midazolam to effect (0.2 mg/kg) and maintained with isoflurane in a mixture of oxygen and medical air. Ringer’s lactate is administered intravenously during the CT exam.

For the study, the population was divided into two groups: group P underwent Dynamic 4D CT (called the perfusion group); group M underwent bolus-activated dual-source, dual-energy multiphase CT (called the multiphase group).

#### 2.2.1. Perfusion Group (Group P)

Unenhanced images of the abdomen were obtained with 0.6 mm collimation to define the craniocaudal extent of the pancreas. A scan acquisition field was set covering all the pancreas with a cranio-caudal width of 22 cm. The dynamic imaging sequence consisted of 33 acquisitions of 0.25 s duration (rotation time) at an interval of 1.5 s (cycle time), resulting in a total examination time of 50 s. Slice thickness was set at 0.6.6 mm, pitch 0.5, and acquisition parameters were set at 200 mAs and 120 kVp (one source-acquisition). The acquisition started with a pre-set delay of 2 s after the injection of contrast medium. Subsequently a whole-body CT scan was performed at a standard delay of 120 s.

#### 2.2.2. Multiphase Group (Group M)

A bolus-triggered arterial phase was acquired after obtaining a 100 HU caudal thoracic aortic enhancement. Scanning parameters were as follows: single-source arterial phase acquisition with a peak voltage of 120 kVp; a tube current-time product of 200–300 mAs with automatic tube current modulation; a slice thickness of 0.6 mm, pitch 0.5, a field of view from the neck down to and including the pelvis. After a 20 s fixed delay, a dual-source, dual-energy portal phase of the abdomen was acquired with one 100 kVp radiation tube and a second 150 kVp tube; other parameters were comparable with the arterial phase.

Detailed specifications of the acquisition parameters for group P and group M are presented in Table 1.

### 2.3. Diagnostic Imaging Analyses

All CT images were retrieved from the Picture Archiving Communication System (PACS) and analyzed using a dedicated stand-alone workstation and vendor-specific post-processing software (SyngoVia, Siemens, Erlangen, Germany). Images were reconstructed using a non-enhancing, non-smoothing reconstruction algorithm with a 512 × 512 matrix size and 50% overlapping section thickness. CT studies were reviewed and analyzed by consensus by two DVMs specialized in diagnostic imaging, with 3 and 2 years of experience in advanced diagnostic imaging, respectively (V.C. and C.C.), under supervision of a DVM with 25 years of experience in advanced diagnostic imaging and a PhD in Radiology (G.B). The radiologists reviewing the studies were aware of the suspected diagnosis of insulinoma. The images were independently evaluated by each radiologist, and their findings were compared. In cases of disagreement, a consensus was reached.

#### 2.3.1. Quantitative Analysis

For quantitative analysis, regions of interest (ROI) were manually positioned in the dedicated anatomical fields on the transverse plane.

For patients in the Dynamic 4D CT group (Group P), four circular regions of interest (ROIs) were drawn: one over each pancreatic lesion, sized according to the lesion’s dimensions; one on the most homogeneous, vessel-free portion of the pancreatic parenchyma; and one each covering the entire cross-section of the cranial abdominal aorta and the portal vein at the level of the hepatic hilum. From the analysis of these four ROIs, the software created four time-attenuation curves (TACs) with the attenuation expressed by Hounsfield units (HU) on the *Y*-axis and the time expressed in seconds (s) on the *X*-axis (Figure 1*).* As previously described [19], four vascular phases were identified: (1) the early arterial phase (EAP), in which only the aorta is enhanced, before the portal vein begins to enhance; (2) the late arterial phase (LAP), after the peak of aortic attenuation when clear inflow of contrast into the portal vein is observed; (3) the pancreatic phase (PP), at the peak of attenuation of normal pancreatic parenchyma; and (4) the portal venous phase (PVP), at the peak of portal vein enhancement. Tumor and parenchymal attenuation (HU) values were recorded at these four time points.

Through the analysis of the attenuation curves of the pancreatic lesions and the normal pancreatic parenchyma, it was also possible to record the time-to-peak (TTP), i.e., the time gap between the administration of the contrast medium and the moment of maximum enhancement of the interested structure.

For patients undergoing dual-source, dual-energy bolus-triggered multiphase CT and thus included in group M, images obtained from (1) the arterial phase, (2) the low-energy tube at 100 KVp, (3) the high-energy tube at 150 KVp, and (4) a third virtual monoenergetic image (VMI) calculated by the software at 120 KeV, were evaluated synchronously in the transverse plane (Figure 2) during the portal phase. Lesions were identified, and a circular ROI was placed on the transverse plane over the pancreatic lesion and over normal pancreatic parenchyma; all attenuation values (HU) were recorded for the arterial phase and for all energy levels of the portal phase (Figure 3).

Quantitative measures of tumor visibility were calculated by analyzing tumor visibility within the context of the pancreatic parenchyma via the tumor-to-pancreas ratio:(1)$$\mathrm{TPR}\;=\;\frac{mean\;HU\;tumor}{mean\;HU\;pancreas}$$

Image noise was assessed by evaluating the contrast-to-noise ratio:(2)$$\mathrm{CNR}\;=\;\frac{mean\;HU\;tumor-mean\;HU\;pancreas}{SD\;HU\;subcutaneous\;tissue}$$These methods were performed using the established tumor-to-pancreas ratio and contrast-to-noise ratio, as reported in the human radiology literature [8].

#### 2.3.2. Qualitative Analysis

For morphological CT analysis, pancreatic lesions were first classified as single or multiple (single/multiple). Lesion size (maximum diameter in mm) was then recorded, considering the largest lesion in cases of multiple lesions. For location (right lobe/body/left lobe), the pancreas was divided into three portions using adjacent anatomical landmarks: the right lobe in the mesoduodenum, the left lobe within the deep wall of the greater omentum, and the body connecting the two lobes. Lesion location was determined based on the site of the largest lesion, and for lesions at the border between the body and a lobe, the location was assigned according to where the majority of the lesion (>50%) was situated (body or lobe). Localization was determined by consensus among the authors in cases of discrepancy. In addition, the following features were recorded: lymph node enlargement (yes/no), lesion-like lymph node enhancement (yes/no), lesion-like liver nodule enhancement suspected of metastasis (yes/no), vascular invasion (yes/no), and vascular encasement (yes/no). In cases where lymph nodes were altered in size or enhanced, they were identified and registered based on their location. Similarly, vessels involved in an invasion or encasement process were recorded. On post-processing analysis, tumor visibility on enhanced images was assessed and classified as visible or not visible (cut-off value of 10 HU difference between the lesion and surrounding parenchyma). Both true non-contrast images and virtual non-contrast images were analyzed depending on their availability and the lesions were classified as hypoattenuating or isoattenuating based on a difference of at least 10 HU. Finally, the operative reports of patients that underwent surgery were retrospectively reviewed, and the definitive location of the lesions was recorded.

### 2.4. Statistical Analysis

Qualitative data were summarized using percentages. Quantitative variables were reported as mean and standard deviation (SD) or as median and interquartile range (IQR), according to the Shapiro–Wilk’s test for normality. Statistical differences between two independent groups were analyzed with Student’s *t* test, under the hypotheses of normality and homoscedasticity, or with the Mann–Whitney rank test for non-normal variables. Nonparametric Friedman tests were used for repeated measures ANOVA, and post hoc analyses were performed through pairwise comparisons between group levels with Holm correction for multiple tests. Associations between qualitative variables were studied with the Chi-square test, while relationships between quantitative variables were measured using Spearman’s correlation index. Linear mixed-effects models assessed differences in TPR and CNR between modalities. The significance level was set at *p* < 0.05. Data were analyzed using statistical software R, version 4.3.2.

## 3. Results

### 3.1. Population

A total of 85 dogs had a final diagnosis of insulinoma between 2016 and 2023, of which 80 underwent CT and 70 met the inclusion criteria. The median age was 10 years (range: 5 to 14.5 years). The mean body weight was 23.21 kg in the P group and 19.51 kg in the M group, with no statistically significant difference between the two groups.

### 3.2. Quantitative Analysis

Among the 70 patients who met the inclusion criteria, 40/70 (57%) underwent Dynamic 4D CT and were included in group P while 30/70 (43%) underwent bolus-triggered, dual-source, dual-energy multiphase CT and were therefore included in group M.

From statistical analysis, both TPR and CNR were statistically significantly higher in the perfusion group (group P) than in the multiphase group (group M) with a *p*-value of 1.09^e−10^ (Figure 4).

In the perfusion group, even if not statistically significant, among the vascular phases analyzed, the late arterial phase was the one with the highest conspicuity both on TPR and CNR (Figure 5).

CT perfusion analysis revealed a tumor’s mean time-to-peak (TTP) of 38.8 s (SD ± +/−5.46 s) and a normal pancreatic parenchyma’s mean TTP of 41.25 s (SD ± 5.79 s).

From the analysis of the conspicuity of the lesions of group M, that is in the arterial phase, portal at 100 kVp, 120 keV, and 150 kVp, there was no significant difference.

Evaluation of true non-contrast images for the perfusion group and virtual non-contrast images for the multi-phase group showed that 21/70 (30%) were hypoattenuating with respect to the surrounding parenchyma whereas 49/70 (70%) of the lesions were isoattenuating and therefore indistinguishable.

### 3.3. Qualitative Analysis

Of the 70 patients included in the study, 21/70 (30%) had histological confirmation of insulinoma and 20/70 (28.6%) had cytological confirmation (obtained in 13/20 from the pancreatic lesion, 3/20 from the lymph nodes, and 4/20 from liver metastases). For the remaining 29/70 patients, the confirmation was considered accurate by the congruence of clinical, laboratory, and imaging findings. The number of lesions was single in 59/70 (84.3%), with the location in the right lobe in 24/70 (34.3%), the pancreatic body in 29/70 (41.4%), and the left lobe in 17/70 (24.3%). The primary lesion size was extremely variable with a mean of 15.5 mm (overall from a minimum of 3 to a maximum of 80 mm), with 15 patients with a lesion < 10 mm. In 24 of 70 lesions (34.3%), the pancreatic contour remained unchanged, indicating that the lesions did not deform the organ’s profile.

In total, 15/70 patients underwent surgery (21.4%) and in 15/15 cases (100%) the location of the pancreatic lesion reported on the tomographic report coincided with the location of the surgical report.

Vascular invasion by the lesion was identified in 14/70 (20%) of patients, in particular involving: 5/14 cranial pancreaticduodenal veins, 4/14 portal veins, 4/14 caudal pancreaticduodenal veins, 1/14 splenic veins, and 1/14 cranial pancreatic-duodenal artery, with one patient showing both portal vein and cranial pancreaticduodenal artery invasion simultaneously. Only two patients showed vascular encasement by the lesion specifically with involvement of the cranial pancreaticduodenal vein.

Hepatic lesions exhibiting an enhancement pattern similar to that of the pancreatic lesions, and therefore considered consistent with metastasis, were identified in 28 out of 70 patients (40%). One or more abdominal lymph nodes were enlarged in 45/70 (64.3%) patients, including: 20/70 left portal, 20/70 right portal, 19/70 jejunal, 13/70 pancreaticduodenal, 12/70 splenic, 2/70 gastric, 1/70 right colic, and 1/70 right renal. In 39/70 (55.7%) lymph node enhancement was synchronous to that of the pancreatic lesion; out of these 39, 38 were increased in size.

## 4. Discussion

This study objectively evaluates the visibility of canine insulinomas through quantitative values of tumor-to-pancreas ratio (TPR) and contrast-to-noise ratio (CNR). Dynamic 4D CT and dual-source, dual-energy multiphase CT acquisition in canine patients with insulinoma have not been described before, and the use of TPR and CNR allowed a comparison between these two acquisition modalities.

In our population, the majority of dogs were medium and large sized with a minority of small and toy sized breeds, in accordance with previous reports. Most patients were crossbreeds, likely reflecting the patient demographic of the institution; among the purebred dogs, the most represented breeds were Boxer, West Highland White Terrier, and notably Golden Retriever, which had not previously been described as highly predisposed. In contrast to a previous report, no statistically significant association was observed between patient age or sex and the condition, although our study group was considerably smaller than that of the epidemiological study by Kraai et al. [2].

As mentioned in the introduction, there are conflicting findings regarding the CT appearance and behavior of canine insulinomas, likely due to the use of different CT scanners and scanning protocols. As suggested by Buishand in 2022 [3], reducing heterogeneity in results requires standardized study designs and the adoption of more accurate imaging techniques, such as the test bolus or bolus tracking methods. In 2023, Skarbek et al. [19] reported that 76% of insulinomas examined using a standardized CT protocol—with dogs in sternal recumbency and a bolus-triggered, quadruple-phase CT scan—were more visible during the late arterial phase. However, in that study, lesion visibility was assessed subjectively using a qualitative grading scale (none, poor, good, excellent) [19].

In human radiology literature, tumor-to-pancreas ratio (TPR) and contrast-to-noise ratio (CNR) have been described to quantitatively assess conspicuity in pancreatic carcinomas [8]. Conspicuity has been described in veterinary literature as a subjective parameter [20,21,22,23], and quantitative CT assessment of visibility has been used to describe nasal conditions [24,25].

The use of TPR and CNR in the present study showed that both tumor-to-pancreas ratio and contrast-to-noise ratio were statistically significantly higher in patients undergoing Dynamic 4D CT than in patients undergoing multiphase DECT, demonstrating objectively better conspicuity of the lesions with Dynamic 4D CT than multiphase DECT. This is likely related to the perfusion protocol itself, which permitted the pancreas to be studied sequentially over time, allowing lesions with transient enhancement to be visualized [11]. In contrast, the multiphasic study provided data at only three time points, and it is possible that, despite bolus triggering, image acquisition did not align with the lesion’s peak enhancement.

The timing at which most insulinomas show their maximum enhancement was 38.8 s (SD ± 5.46 s), just before the peak of the pancreatic parenchyma. These findings indicate that most insulinomas enhance between 34 and 44 s following contrast injection into a cephalic vein, representing a narrower enhancement window than that reported by Skarbek in 2023 [19], which ranged from 30 to 54 s. Consistent with previous veterinary literature [19], maximum conspicuity of the lesion occurred during the late arterial phase of the dynamic 4D perfusion studies. Although this finding was not statistically significant, it remains clinically relevant, confirming that the late arterial phase is the period in which insulinomas are most conspicuous and best characterized. This information allows for optimized timing of image acquisition in suspected insulinoma cases, improving lesion detectability even with lower-performance CT scanners and standard-delay protocols.

Dual-energy CT analysis did not reveal any statistically significant results in this study. At our center, the abdominal CT protocol includes DE scanning in the portal venous phase, whereas the arterial and interstitial phases are performed using single-energy acquisition (120 kVp). Three different CT volume datasets from dual-energy (DE) scans are routinely evaluated: images at 100 kVp, 120 keV (mixed), and 150 kVp. In humans, for general abdominal applications, virtual monoenergetic images at 70 keV have been demonstrated to have better objective and subjective image quality than at 120 kVp images [26]. A lower energy level is often used for pancreatic applications, with most reported studies using 50–70 keV VMI images as optimal for pancreatic imaging, as these lower energy levels accentuate subtle differences in iodine contrast [13]. Another study, with both arterial and portal phases acquired with dual-energy spectral imaging, reported an optimal energy level based on CNR measurement for pancreatic tumors in the range of 45–60 keV [27]. Ideally, both acquisition and post-processed VMI at lower energy levels (50–100 keV) would have been evaluated. However, the resulting increase in image noise may have compromised the assessment of other abdominal structures—crucial for detecting metastases and guiding informed treatment decisions [3].

In non-contrast scans, both TNC and VNC, most insulinomas were isoattenuating to the surrounding parenchyma, in agreement with previous studies [4,17]. Considering that 70% of insulinomas were isoattenuating to the surrounding pancreas on non-contrast scans, pre-contrast acquisitions become of questionable utility in dogs with suspected insulinoma, since removing direct scanning from the protocol would reduce acquisition time and radiation exposure.

As previously reported [17,18,26,28,29], the majority of patients in our study (~84%) presented with a single lesion. In the present study, 41.4% of the lesions were located in the pancreatic body, in contrast to what is reported in the literature, where the majority of lesions involve the pancreatic lobes rather than the body [29]. Regarding the size of insulinomas, there are currently no studies in the veterinary literature that specifically investigate the relationship between tumor size and lesion visibility. Existing studies have only explored the correlation between tumor size and prognosis, which appears to be non-significant [29]. In the present study, the primary lesion was identified in 100% of patients, regardless of size. Among the 70 patients, 15 had lesions smaller than 10 mm, and in 34% of patients, the pancreatic lesions did not deform the organ’s profile.

In contrast to previous literature [3], the present study demonstrated that standardization of CT studies—including optimal patient positioning, appropriate CT scan and contrast injection protocols, and post-processing analyses—allowed for the confirmation of one or more nodules and accurate localization in all patients, regardless of lesion size.

Regarding vascular involvement of insulinomas, veterinary literature provides little information. In the present study, 20% of patients showed vascular invasion, predominantly in the cranial pancreaticduodenal vein and portal vein, mostly depending on tumor localization. Less frequently involved vessels were the caudal pancreaticduodenal vein and cranial pancreaticduodenal artery. Only two cases showed vascular encasement of the cranial pancreaticduodenal vein.

It is reported that at the time of diagnosis approximately 50% of canine insulinomas have metastases to regional lymph nodes or liver [3].

In agreement with the previous literature, at the time of examination, 40% of patients included here had liver lesions with synchronous enhancement to the primary lesion indicative of metastases [19], in four cases confirmed cytologically.

Regarding lymph nodes, in this study ~55% of patients showed one or more regional lymph nodes enhancing synchronously with the primary lesion, which could raise suspicion of metastasis even in the absence of enlargement [19]. However, cytological confirmation was available in only three cases, rendering these data of limited significance and highlighting that the potential metastatic behavior of lymph nodes in insulinoma remains a matter for future studies, supported by cytological or histological confirmation.

## 5. Conclusions

This study showed that dynamic perfusion CT evaluation provided significantly higher lesion conspicuity than multiphase CT for detecting canine insulinomas, with maximum tumor enhancement occurring between 34 and 44 s after contrast injection, corresponding to a mean time-to-peak (TTP) of 38.8 s, just prior to the peak enhancement of the pancreatic parenchyma and aligning with the late arterial phase of perfusion CT.

## Figures and Tables

**Figure 1 vetsci-12-01102-f001:**
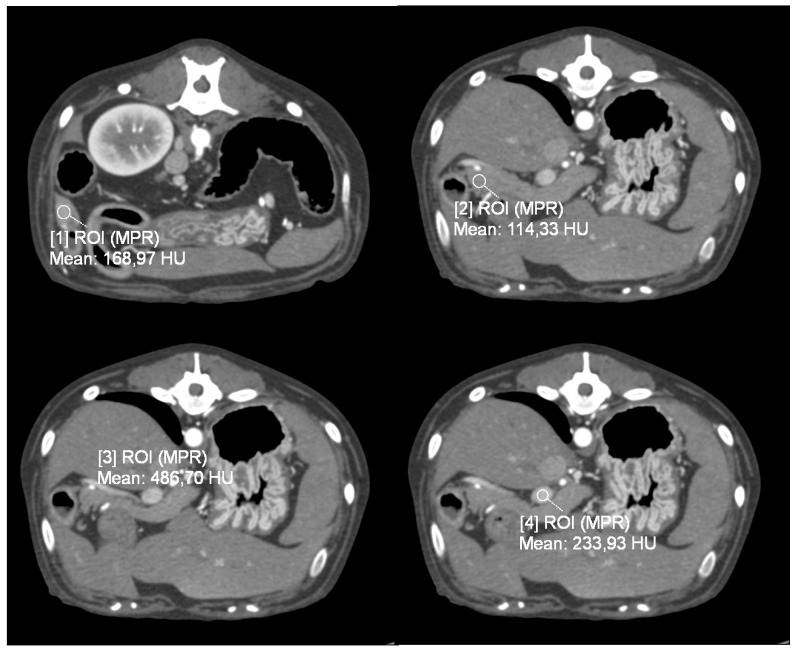
Transverse plane, images from dynamic 4D CT data; WW 700, WL 80. A region of interest (ROI) was placed over the identified pancreatic lesion, normal pancreatic parenchyma, cranial abdominal aorta, and portal vein.

**Figure 2 vetsci-12-01102-f002:**
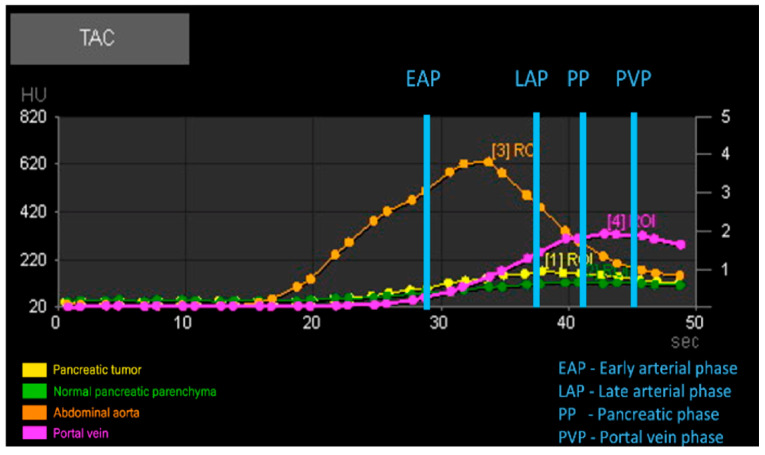
Example of the automatic output generated by perfusion analysis of the regions of interest (ROIs) with graphical representation of time attenuation curves (TACs) and vascular phases.

**Figure 3 vetsci-12-01102-f003:**
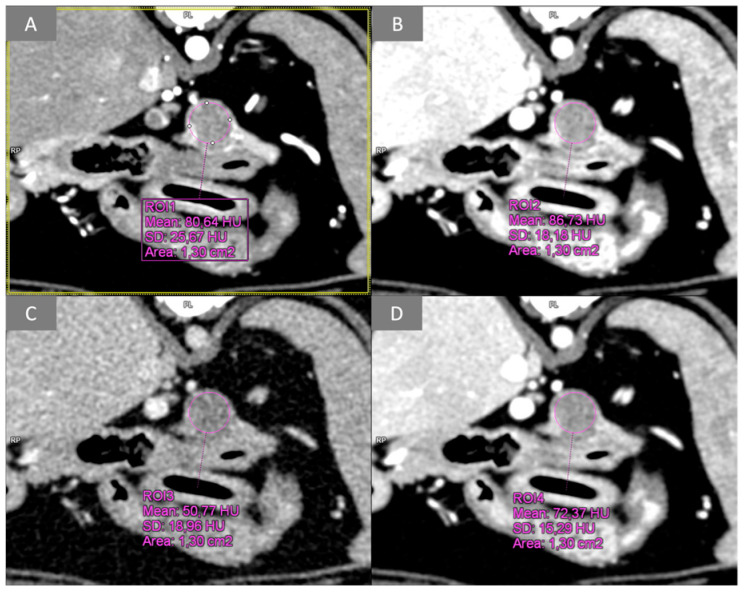
Transverse plane, images from multiphase dual-energy CT data; WW 400, WL 40. Example of pancreatic lesion ROI analysis during the arterial phase and at different energy levels of the portal phase: (**A**) arterial phase; (**B**) low energy PVP (100 kVp); (**C**) high energy PVP (150 kVp); (**D**) VMI PVP (120 keV). PVP, portal venous phase; VMI, virtual monoenergetic image.

**Figure 4 vetsci-12-01102-f004:**
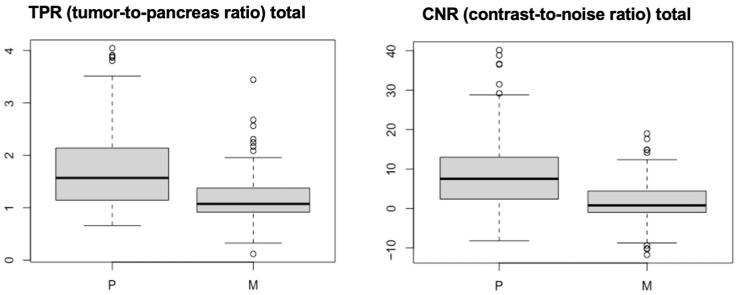
Boxplots of TPR and CNR for groups P and M. The mean values of both measures are significantly higher in group P than in group M.

**Figure 5 vetsci-12-01102-f005:**
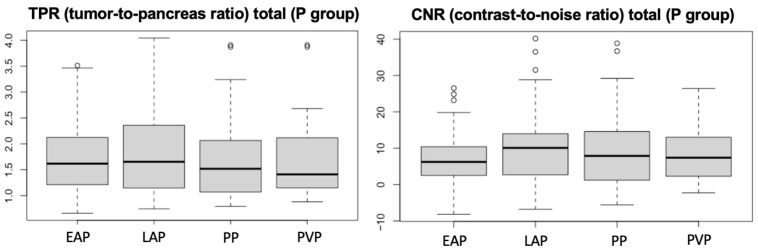
Boxplots of TPR and CNR for group P, with statistical analysis indicating the mean values. EAP= early arterial phase, LAP = late arterial phase, PP = pancreatic phase, PVP = portal venous phase.

**Table 1 vetsci-12-01102-t001:** Acquisition parameters for the two study groups: group P (Dynamic 4D CT, perfusion group) and group M (bolus-activated arterial phase and dual-source, dual-energy multiphase CT, multiphase group).

Scan Mode	Arterial Phase	Dual-Energy Portal Venous Phase	Dynamic 4D CT
	(Group M)	(Group M)	(Group P)
Scanner	SOMATOM Force	SOMATOM Force	SOMATOM Force
Tube current (mAs)	600	400–800	220
Tube voltage (kV)	120	100–150	120
Slices	0.66 mm (Acq. 192 × 0.66 mm)	0.66 mm (Acq. 192 × 0.66 mm)	0.6 mm (Acq. 192 × 0.66 mm)
Direction	Cranio-caudal	Cranio-caudal	Caudo-cranial
Rotation time	0.5 s	1 s	0.33 s
Pitch	0.6	0.5	0.5
Reconstruction kernel	Br40	Br40	Bv36
CT Dose Index (mGy)	15–48	13–43	474–788
Cycle time	—	—	1.5 s
Position increment	—	—	0.3
Field of view (FOV)	—	—	200 × 200 mm

## Data Availability

The original contributions presented in this study are included in the article. Further inquiries can be directed to the corresponding authors.

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
