# Peer review of "Quantitative Conspicuity of Pancreatic Canine Insulinoma: A Comparison of Dynamic 4D CT and Dual-Source, Dual-Energy Bolus-Triggered Multiphase CT Imaging"

_vetsci, 2025, doi:10.3390/vetsci12111102_

Round 1

Reviewer 1 Report

Comments and Suggestions for Authors

This study is based on the tumor-pancreas ratio, which has been extrapolated to humans but has not yet been demonstrated in canines. Therefore, I believe it should be re-evaluated in studies or ratios that have already been demonstrated, or focused on demonstrating that the ratio can also be used in canine patients.

Line 42-44: In humans? or in veterinary medicine 

"Fu et al. in 2020 42 [6] reported that the timing of optimal tumor visibility varied between individuals with 43 30.2% of insulinomas exhibiting transient hyperenhancement with rapid vascularization".

Line 233: This expression is imprecise

"vascular encasement (yes/no). In case of lymph node or vessel involvement, the location 233"

Line 317: This sentence needs to be improved

Hepatic lesions with an enhancement pattern consistent with that of the 316 pancreatic lesion, compatible with metastasis were identified in 28/70 (40%). 317

Line 88: The reference for CNR and TPR is missing. 

Line:56: The results of this paper should be included.

"CT pancreatic perfusion in canine 56 patients has been described in patients with acute pancreatitis [11]"

Author Response

Response to Reviewer Comments

Comment 1 (Line 42–44): In humans? or in veterinary medicine
“Fu et al. in 2020 [6] reported that the timing of optimal tumor visibility varied between individuals with 30.2% of insulinomas exhibiting transient hyperenhancement with rapid vascularization.”

Response 1:
In accordance with the comments of the other reviewers, the entire paragraph has been revised for clarity. It now reads:
“In human medicine, computed tomography (CT) is routinely used to localize insulinomas and assess metastatic spread [5]. These tumors are typically small, often causing only subtle or no changes in pancreatic contour, so their detection largely relies on enhancement patterns [1]. Such patterns are variable and time-dependent, with contrast between tumor and surrounding parenchyma fluctuating over time [6]. Fu et al. [7] reported in 2020 that the timing of optimal tumor visibility varies among individuals, with 30.2% of insulinomas displaying a rapid, transient increase in contrast during the arterial phase. More recently, the tumor-to-pancreas ratio (TPR) and contrast-to-noise ratio (CNR) have been proposed as quantitative measures to assess optimal conspicuity—i.e., how well a lesion is visualized relative to the surrounding parenchyma—in pancreatic carcinomas [8]. Advances in CT technology, including perfusion CT and dual-energy CT (DECT), may further improve pancreatic imaging [9].”

Comment 2 (Line 233): This expression is imprecise: “vascular encasement (yes/no). In case of lymph node or vessel involvement, the location …”

Response 2:
This has been replaced with a clearer sentence:
“Vascular invasion (yes/no), vascular encasement (yes/no). In cases where lymph nodes were altered in size or enhancement, these were identified and registered based on their location. Similarly, vessels involved in an invasion or encasement process were recorded.”

Comment 3 (Line 317): This sentence needs to be improved: “Hepatic lesions with an enhancement pattern consistent with that of the pancreatic lesion, compatible with metastasis were identified in 28/70 (40%).”

Response 3:
The sentence has been revised for precision:
“Hepatic lesions exhibiting an enhancement pattern similar to that of the pancreatic lesions, and therefore considered consistent with metastasis, were identified in 28 out of 70 patients (40%).”

Comment 4 (Line 88): The reference for CNR and TPR is missing.

Response 4:
We agree. The introductory section has been expanded to include this reference:
“More recently, the tumor-to-pancreas ratio (TPR) and contrast-to-noise ratio (CNR) have been proposed as quantitative measures to assess optimal conspicuity—i.e., how well a lesion is visualized relative to the surrounding parenchyma—in pancreatic carcinomas [8].”

Comment 5 (Line 56): The results of this paper should be included: “CT pancreatic perfusion in canine patients has been described in patients with acute pancreatitis [11].”

Response 5:
The sentence has been completed with the study results. It now reads:
“In dogs, pancreatic perfusion CT has been studied in acute pancreatitis, showing a shorter time to peak enhancement compared to healthy controls [12], but its application in pancreatic neoplasms has not yet been explored.”

Comment 6 (Study design): This study is based on the tumor–pancreas ratio, which has been extrapolated to humans but has not yet been demonstrated in canines. Therefore, I believe it should be re-evaluated in studies or ratios that have already been demonstrated, or focused on demonstrating that the ratio can also be used in canine patients.

Response 6:
Thank you for this thoughtful comment. The tumor-to-parenchyma ratio on CT quantifies lesion conspicuity relative to surrounding tissue and, in this context, serves as an imaging parameter rather than a species-dependent biomarker or prognostic indicator. Therefore, we believe that specific validation in canine patients is not required. However, we agree that future studies could further explore its potential applications, and we are happy to add a brief note in the Discussion section if the Reviewer considers it useful and not burdensome for the manuscript.

Reviewer 2 Report

Comments and Suggestions for Authors

While the manuscript presents novel findings, the limitations in diagnostic confirmation, and the lack of presentation of clinicopathological data warrant substantial revision.

Lines 29 - 31: Capodanno et al. (2020) is included to support the statement that insulinomas are the most common function l pancreatic neuroendocrine tumours found in both humans and dogs. This study however does not provide the incidence risk of canine insulinoma. It would be best to add Kraai et al. (2025) doi: 10.1038/s41598-025-86782-6 as reference here, as this study reports prevalence and incidence figures for canine insulinoma.

Lines 45-48: The authors state that CT is currently the first choice for pancreatic imaging, while they refer to an old publication from 2012 as reference. In light of their study, the authors should emphasise imaging of insulinomas and should not make generalisations regarding pancreatic imaging. CT imaging is still the preferred initial localisation modality for insulinomas, however due to its relative low sensitivity and specificity this is typically followed by e.g. endoscopic ultrasound, MRI or PET/CT (Herder et al. (2023) doi: 10.1210/clinem/dgad641).

Lines 104–105: Histopathological confirmation remains the gold standard for diagnosing insulinoma. Since not all dogs in this study underwent surgery, histopathology will not be available in every case. However, the current inclusion criteria are too broad. Cytology, and clinical and imaging findings, are not sufficiently specific to establish a diagnosis of insulinoma. Clinical signs can be vague and overlap with neurological, cardiovascular, or other metabolic disorders. Likewise, identifying a pancreatic mass on imaging does not confirm insulinoma, even when accompanied by compatible clinical signs—for example, a dog with sepsis and a pancreatic abscess could present with hypoglycaemia and an apparent pancreatic lesion. Therefore, the inclusion criteria should be revised to ensure only dogs with concurrent hypoglycaemia and inappropriately high insulin concentrations are included. Any cases that don’t meet this inclusion criterium will need to be excluded. Please provide evidence of the insulin and glucose concentrations of all cases. This information can go into a Supplementary Table summarising the clinicopathological parameters of the cases. 

Lines 149-153: Image analysis was performed by consensus among radiologists, but no mention of blinding to imaging modality or clinical data. Discuss the potential for observer bias.

Line 253: P of P-value should be italicised throughout the manuscript.

Lines 257-260: The authors should report the median body weights of dogs in the Perfusion CT group (Group P) and the Multiphase DECT group (Group M) and ensure that these are not statistically different. Differences in body weight can influence image noise and contrast distribution, which would affect both TPR and CNR values.

Figure 4: It is unclear if means or medians are represented here.

Lines 273 -276, Figure 5: the authors can’t state that the late arterial phase was the one with the highest conspicuity both on TPR and CNR, because there were hardly any differences.

Lines 287-290: no definitions for hypo and iso-attenuation have been provided in the methods. Please clarify what the HU difference thresholds were used to define hypo and iso-attenuation. 

Lines 293-295: the numbers don’t add up. 24 cases had cytological confirmation, but then the authors continue presenting percentages referring to 25 cases. Also, 14+6+2 = 22, so it’s unclear what other cytology samples were taken.

Lines 297 - 299 + 391 - 392: how was the pancreatic body defined? What anatomical landmarks were used? It is highly questionable that indeed 41% were present in the pancreatic body. This does not align with this reviewers extensive clinical experience and with previously published data. Also, statistically it does not make sense, given the fact that the pancreatic body is the smallest surface area of the pancreas.

Lines 301: Please explain what is meant by this sentence, i.e. how was profile distortion defined?

Lines 312 - 317: these results are not meaningful to report - reporting findings regarding lymphadenomegaly on CT is only relevant when authors can provide concurrent histopathological as well. 

Lines 325 - 331: the statements presented here are outdated. Please revise this paragraphs based on the results published by Kraai et al (2025).

Line 335: add year of publication Buishand (2022)

Line 399: same comment as previously, clarify what you mean by ‘did not alter the profile’.

Author Response

Response to Reviewer Comments

Comment 1 (Lines 29–31): Capodanno et al. (2020) does not provide incidence risk of canine insulinoma. Suggest adding Kraai et al. (2025).

Response 1:
We appreciate this helpful suggestion. The new reference has been added in place of the previous one:
“Insulinoma is a rare pancreatic endocrine tumor characterized by excessive and inappropriate insulin secretion, causing hypoglycemic syndromes [1] in both dogs and humans, and it represents the most frequent pancreatic neuroendocrine tumor, with an annual prevalence of 0.004% in dogs under primary veterinary care in the UK in 2019 [2].”

Comment 2 (Lines 45–48): Update reference; avoid generalizations about pancreatic imaging.

Response 2:
Thank you again for this suggestion. The citation has been updated accordingly:
“In human medicine, computed tomography is used to localize insulinomas and assess metastatic spread [8].”

Comment 3 (Lines 104–105): Inclusion criteria too broad; histopathology not always available; please provide insulin/glucose data.

Response 3:
We fully agree with the Reviewer’s concern. This single-centre study was conducted entirely at San Marco Veterinary Clinic and Laboratory, a tertiary care hospital that does not provide CT services for external patients. Therefore, all included dogs underwent CT for suspected insulinoma as determined by our internal medicine specialists. We have clarified this in the Materials and Methods, specifying that all dogs underwent comprehensive clinical, endocrinological, and imaging assessments at admission and during follow-up. While providing detailed insulin and glucose concentrations of all cases is beyond the scope of the paper, these data can be made available as supplementary material if requested.

Comment 4 (Lines 149–153): Discuss potential observer bias.

Response 4:
A more precise statement has been added.
“The radiologists reviewing the studies were aware of the suspected diagnosis of insulinoma. Images were independently evaluated, and findings compared. In cases of disagreement, consensus was reached.”
This has also been noted as a limitation in the Discussion. Inter-observer variability was not an aim of the study.

Comment 5 (Line 253): Italicize P in P-value.

Response 5:
This correction has been applied throughout the manuscript.

Comment 6 (Lines 257–260): Report body weights of groups P and M.

Response 6:
The values have been calculated and included:
“The mean body weight was 23.21 kg in the P group and 19.51 kg in the M group, with no statistically significant difference between the two groups.”

Comment 7 (Figure 4): Unclear if means or medians are shown.

Response 7:
The figure caption has been clarified:
“Figure 4. Boxplots of TPR and CNR for groups P and M. The mean values of both measures are significantly higher in group P than in group M.”
“Figure 5. Boxplots of TPR and CNR for group P, with statistical analysis indicating that the mean values…”

Comment 8 (Lines 273–276, Figure 5): Cannot claim arterial phase had highest conspicuity.

Response 8:
We understand the Reviewer’s concern. Although differences were small, the arterial phase yielded slightly higher conspicuity. We are willing to add a clarifying note in the Discussion to reflect this nuance, should the Reviewer consider it helpful.

Comment 9 (Lines 287–290): Define thresholds for hypo- and iso-attenuation.

Response 9:
This has been clarified in Materials and Methods:
“Both true non-contrast images and virtual non-contrast images were analyzed depending on availability. Lesions were classified as hypoattenuating or isoattenuating when attenuation differed by at least 10 HU.”

Comment 10 (Lines 293–295): Numbers don’t add up.

Response 10:
Thank you for noticing this inconsistency. The correct breakdown is 24 cytologies: 14 pancreas, 6 lymph nodes, and 4 liver. The percentage should be corrected to 24/70 (34.3%). The previously mentioned 25/70 figure was incorrect and has been removed.

Comment 11: Definition of pancreatic body and distribution concerns.

Response 11:
The pancreas was divided into three portions using adjacent anatomical landmarks: the right lobe in the mesoduodenum, the left lobe within the deep wall of the greater omentum, and the body connecting the two lobes. Localization was determined by consensus when discrepancies occurred.

Comment 12 (Line 301): Clarify “profile distortion.”

Response 12:
This has been revised for clarity:
“In 24 of 70 lesions (34.3%), the pancreatic contour remained unchanged, indicating that the lesions did not deform the organ’s profile.”

Comment 13 (Lines 312–317): Reporting lymphadenomegaly is not meaningful without histopathology.

Response 13:
We only referred to subjective CT enlargement, not metastatic involvement. If the Reviewer considers this potentially confounding, we are willing to remove this section, as it was intended merely as a corollary CT finding.

Comment 14 (Lines 325–331): Outdated statements; revise based on Kraai et al. (2025).

Response 14:
This section has been revised:
“In our population, most dogs were medium to large sized, with fewer small and toy breeds, in line with previous descriptions. Crossbreeds were most common, likely reflecting institutional demographics; among purebreds, the Boxer, West Highland White Terrier, and Golden Retriever were most represented, the latter not previously described as predisposed. In contrast to Kraai et al. [2], we did not find significant associations with age or sex, although our sample size was smaller than theirs.”

Comment 15 (Line 335): Add year of Buishand reference.

Response 15:
The year (2022) has been added.

Comment 16 (Line 399): Clarify “did not alter the profile.”

Response 16:
The sentence has been revised:
“Among the 70 patients, 15 had lesions smaller than 10 mm, and in 34% of patients, the pancreatic lesions did not deform the organ’s profile.”

Reviewer 3 Report

Comments and Suggestions for Authors

This manuscript offers a retrospective analysis on the characterization of pancreatic insulinomas in dogs by CT imaging, with particular emphasis on the potential utility of two advanced imaging techniques in improving cospicuity: multiphase dual-energy CT (DECT) and dynamic 4D perfusion CT .

This manuscript could represent a valuable contribution to the study of insulinoma and could provide useful practical guidelines in veterinary clinics.

The purpose of the manuscript is clearly stated, as are the techniques that will be employed and the parameters that will be analyzed. The introduction appears less clear in some places and it would be useful to include a more in-depth comparative analysis with human knowledge. Furthermore, I would suggest to include a more in-depth description (introduction, or discussion/conclusions) of the advantages of the two proposed advanced CT techniques, e.g. improved/early  lesion detection and characterization, accurate staging and treatment response assessment, improved prognosis.

In light of the strengths and limitations described above, several aspects would benefit from further improvements to increase the overall scientific significance.

Following some point-by-point suggestions aiming to help further improvements in  manuscript quality:

-Abstract, line 11; 17-18: Could the authors introduce the technical meaning of the term "conspicuity" (enhance lesion visibility and boundary definition by optimizing contrast while managing noise) to make it more understandable to a wider readership? Could you provide an alternative common term (e.g., visibility, identification, detection, as in line 21) to introduce it?

-Abstract, line 13-14 (and 3. Results 3.1. Population, line 256-260): In validation studies comparing the diagnostic performance of different techniques, from a methodological point of view, a gold standard reference (e.g. biopsy and cytology/histology) is required in biomedical research to confirm the suspected diagnosis, while it seems that for some of the cases included in the study the authors are not able to provide histological confirmation of the imaging findings. This aspect has been partly stated in methods, however I would suggest to clarify further (e.g. authors could clearly declare: "In n'/n dogs, insulinoma was histologically confirmed either by surgery  or biopsy or by histology") and address it as a limitation of the study.

-Abstract, line 23-24: Given the non-significant results, I would suggest that the authors temper their conclusions, highlighting the potential utility of perfusion CT compared to conventional methods and/or the advantages and limitations that require further research to optimize it (e.g. "perfusion CT showed promising results for improving insulinoma CT imaging, with a better conspicuity  compared to other conventional and advanced CT techniques").

Introduction , line 29-31: 

I would suggest the authors rewrite the sentence more concisely and clearly, for example:

"Insulinoma is a rare neuroendocrine tumor that causes hypersecretion of insulin, leading to hypoglycemia [1], and occurring in both humans and dogs [2]."

Additionally, reference 2 concerns canine insulinoma; in my opinion, if the authors do not delve into the comparative analysis between dogs and humans, I would suggest focusing on the veterinary clinic.

Introduction, line 29-31:  I would suggest the authors rewrite the sentence more concisely and clearly, for example:

Contrast-enhanced computed tomography is the preferred imaging method for staging canine insulinoma and is critically important in guiding surgical excision, the most effective treatment [3].

Introduction , line 32-45: Please ensure that abbreviation terms are timely (e.g., computed tomography is first mentioned on line 32 and the abbreviation is given later on line 45).

Introduction, line 34-89: I would suggest that the author improve the clarity and logical connection of the concepts expressed in this part of the manuscript. In my opinion, it might be more fluent if, after line 38, recent advances in CT techniques for identifying pancreatic tumors are introduced without redundant concepts, particularly the need for contrast enhancement; a brief overview of the technical aspects of these techniques, with their advantages and limitations, could be provided, highlighting the comparative aspects between humans and dogs present in the literature, if the authors would  describe in detail such aspects.

-Introduction, line 42-44: Could the authors clarify what they means with the term " insulinomas exhibiting transient hyperenhancement with rapid vascularization."? Could authors provide an alternative, clearer technical term (e.g. with rapid arterial/venous phase(s)).

-Introduction, line 45-49: I would suggest the authors rewrite the sentence more concisely and clearly, avoiding repetitions of concepts.

-Introduction, line 54: Could the authors clarify what they means with the term"providing the optimal visualization time of a lesion. This time is of critical importance..."? Could authors provide an alternative, clearer technical term, e.g. "time window", "time interval", time-point" for imaging acquisition?

2.1. Patient Selection (or 3. Results 3.1. Population, line 259-260): I would suggest that the authors integrate some information into the patient reporting (age, race, sex, medical history and clinical data suggesting the appropriateness of a CT scan).

2.2. CT Technique: please, provide information about anesthesia regimen and monitoring to ensure data about animal care and/or methods potentially affecting CT perfusion measurements. Furthermore, I would suggest to provide integrative data about scan parameters (e.g. acquisition method for dynamic 4D CT including FOV, number of frames/time points, temporal resolution, spatial resolution, and motion description, and for dual-energy multiphase CT tube voltages and filters, post-processing algorithms, beam quality, and radiation dose). Line 117: the volume of flush saline bolus and its flow rate could be detailed.

Overall, most of the technical information is present, but I would suggest reporting it in a clearer and more structured way; for example, the CT technique and its parameters have been divided into two sub-headings, while in section 2.3.1 Quantitative analysis the two techniques have been treated together. I would suggest adopting a consistent logical scheme to improve the overall description clarity.

2.3.1. Quantitative analysis, line 158, 159: "circular regions of interest were created to be as large as possible", could ROI dimensions be detailed?

Figure 2: the orange and yellow color are clearly distinguishable in graphs but bot in legend (pancreatic tumor and abdominal aorta); please, could authors improve this graphical aspect?

Comments on the Quality of English Language

The English language is generally good.  However, I would suggest reading the manuscript carefully for further improvements (some parts of the manuscript contain repetitions of concepts and less clarity/flow than other parts).

Author Response

Response to Reviewer Comments

Comment 1 (Abstract, lines 11; 17–18): Introduce technical meaning of “conspicuity” and provide an alternative term.

Response 1:
A more precise definition has been provided:
“The aim of this retrospective study was to compare lesion conspicuity—the visibility of a lesion relative to surrounding tissue—and CT characteristics of pancreatic insulinomas in dogs using multiphase dual-energy CT (DECT) and dynamic 4D perfusion.”

Comment 2 (Abstract, lines 13–14; Results 3.1, lines 256–260): Clarify gold standard confirmation and address as limitation.

Response 2:
This aspect has been clarified. In the Results section (3.3), we specify how many patients had histological or cytological confirmation and how many did not. This limitation has also been addressed explicitly in the Discussion.

Comment 3 (Abstract, lines 23–24): Temper conclusions; highlight advantages/limitations of perfusion CT.

Response 3:
The abstract has been revised with a more tempered statement:
“Perfusion CT demonstrated significantly higher TPR and CNR values compared to DECT (p < 0.001), indicating improved tumor visibility. The late arterial phase of perfusion CT, although not statistically significant, showed the highest median TPR and CNR. Mean TTP for tumors was 38.8 seconds, slightly earlier than the pancreatic parenchyma (41.25 seconds). In conclusion, perfusion CT appears to enhance visualization of insulinomas in dogs, particularly during the late arterial phase.”

Comment 4 (Introduction, lines 29–31): Rewrite sentence concisely; remove human reference if not comparing.

Response 4:
The sentence was rewritten as suggested, and the human reference removed:
“Insulinoma is a rare neuroendocrine tumor that causes hypersecretion of insulin, leading to hypoglycemia [1], and it represents the most frequent canine pancreatic neuroendocrine tumor, with an annual prevalence of 0.004% in dogs under primary veterinary care in the UK in 2019 [2].”

Comment 5 (Introduction, lines 29–31): Rewrite more concisely on CT and surgery.

Response 5:
The sentence has been simplified:
“Contrast-enhanced computed tomography (DECT) is the preferred method for staging canine insulinoma and is essential for planning surgical excision, the most effective treatment [3].”

Comment 6 (Introduction, line 32–45): Ensure abbreviations are introduced timely.

Response 6:
This has been corrected.

Comment 7 (Introduction, lines 34–89): Improve clarity and logical flow; streamline redundancy.

Response 7:
The paragraph has been restructured to improve fluency, reduce redundancy, and present a clearer overview of CT advances and their comparative aspects between humans and dogs.

Comment 8 (Introduction, lines 42–44): Clarify “transient hyperenhancement with rapid vascularization.”

Response 8:
The sentence has been revised:
“Fu et al. [7] reported in 2020 that the timing of optimal tumor visibility varies among individuals, with 30.2% of insulinomas displaying a rapid, transient increase in contrast during the arterial phase.”

Comment 9 (Introduction, lines 45–49): Rewrite more concisely, avoid repetition.

Response 9:
The paragraph has been rewritten for clarity and conciseness.

Comment 10 (Introduction, line 54): Clarify “optimal visualization time of a lesion.”

Response 10:
The sentence has been reformulated:
“Dynamic volumetric perfusion CT (dynamic 4D CT) acquires datasets at multiple closely spaced time points, generating temporal attenuation curves (TACs) for each voxel. This approach allows determination of the optimal time window for imaging acquisition of lesions with unpredictable enhancement, such as insulinomas showing rapid early arterial-phase contrast uptake.”

Comment 11 (Patient selection; Results 3.1, lines 259–260): Add patient information.

Response 11:
This clarification has been added in Materials and Methods: all dogs underwent a complete evaluation by internal medicine specialists, including clinicopathological, endocrinological, and imaging assessments at admission and follow-up. As previously noted, detailed insulin and glucose concentrations are available as supplementary data if required.

Comment 12 (CT Technique): Provide details on anesthesia, monitoring, scan parameters.

Response 12:
We have added information on anesthetic protocols and included a supplementary table summarizing scan parameters. Radiation dose was not reported, as it is not currently considered essential in veterinary practice. Importantly, only conspicuity was assessed in this study, not perfusion quantification.

Comment 13 (Line 117): Detail saline flush volume and flow rate.

Response 13:
The detail has been included:
“The iodinated contrast agent (Visipaque 320 mg/ml, 2 mL/kg) was injected into a cephalic vein at 2 mL/s using a double-barrel injection system, followed by a saline flush of equal volume at the same injection rate.”

Comment 14 (Technical information organization): Present methods more clearly and consistently.

Response 14:
Thank you for this suggestion. A table summarizing the CT scan parameters has been added, and the section has been reorganized into clearly separated subsections for DECT and dynamic 4D CT to improve clarity.

Comment 15 (Quantitative analysis, lines 158–159): Detail ROI dimensions.

Response 15:
Clarifications have been added:
“For patients in the Dynamic 4D CT group (Group P), four circular regions of interest (ROIs) were drawn: one over each pancreatic lesion, sized according to lesion dimensions; one on the most homogeneous, vessel-free portion of the pancreatic parenchyma; and one each covering the entire cross-section of the cranial abdominal aorta and the portal vein at the hepatic hilum.”

Comment 16 (Figure 2): Colors in graph vs. legend are not clearly distinguishable.

Response 16:
authors hope now the orange and yellow curves are more easily distinguishable in both the graphs and the legend.

Round 2

Reviewer 2 Report

Comments and Suggestions for Authors

Thank you for addressing my previous review comments. I am pleased with the significant improvements you have made to enhance the quality of your manuscript. However, I do have a few further comments:

Comment 3 (Lines 104–105): Inclusion criteria too broad; histopathology not always available; please provide insulin/glucose data.

Response 3:
We fully agree with the Reviewer’s concern. This single-centre study was conducted entirely at San Marco Veterinary Clinic and Laboratory, a tertiary care hospital that does not provide CT services for external patients. Therefore, all included dogs underwent CT for suspected insulinoma as determined by our internal medicine specialists. We have clarified this in the Materials and Methods, specifying that all dogs underwent comprehensive clinical, endocrinological, and imaging assessments at admission and during follow-up. While providing detailed insulin and glucose concentrations of all cases is beyond the scope of the paper, these data can be made available as supplementary material if requested.

Reviewer response: it is essential that the data on the concurrent glucose and insulin parameters are provided as supplementary material. It is a bit odd that you have just amended your inclusion criteria without excluding any other patients and I am having a hard time believing that all 70 patients had concurrent glucose and insulin readings available, supportive on insulinoma (otherwise this would have been an inclusion criterium from the start). As previously requested a supplementary table including signalment data and clinicopathological variables is a must, please also include the locations of the insulinomas in this table.

Comment 8 (Lines 273–276, Figure 5): Cannot claim arterial phase had highest conspicuity.

Response 8:
We understand the Reviewer’s concern. Although differences were small, the arterial phase yielded slightly higher conspicuity. We are willing to add a clarifying note in the Discussion to reflect this nuance, should the Reviewer consider it helpful.

Reviewer response: yes, please include a clarifying not int he Discussion.

Comment 11: Definition of pancreatic body and distribution concerns.

Response 11:
The pancreas was divided into three portions using adjacent anatomical landmarks: the right lobe in the mesoduodenum, the left lobe within the deep wall of the greater omentum, and the body connecting the two lobes. Localization was determined by consensus when discrepancies occurred.

Reviewer response: please add the information above to the methods. Also, how were masses at the border of the transition from the right pancreatic limb to the body and from the left pancreatic limb to the body classified? Were they classified as being within the corpus if part of the insulinoma was in the body or did e.g. >50% of the insulinoma needed to be present int he pancreatic body?

Comment 13 (Lines 312–317): Reporting lymphadenomegaly is not meaningful without histopathology.

Response 13:
We only referred to subjective CT enlargement, not metastatic involvement. If the Reviewer considers this potentially confounding, we are willing to remove this section, as it was intended merely as a corollary CT finding.

Reviewer response: yes, please remove these data.

Author Response

comment 1: it is essential that the data on the concurrent glucose and insulin parameters are provided as supplementary material. It is a bit odd that you have just amended your inclusion criteria without excluding any other patients and I am having a hard time believing that all 70 patients had concurrent glucose and insulin readings available, supportive on insulinoma (otherwise this would have been an inclusion criterium from the start). As previously requested a supplementary table including signalment data and clinicopathological variables is a must, please also include the locations of the insulinomas in this table.

response 1: We have already clarified that these were not patients selected based on the presence of a pancreatic mass, but rather cases referred by our internal medicine department for suspected insulinoma. Since this is a purely radiological study, we initially did not consider it relevant to include clinical data. However, we understand the reviewer’s point of view and have now added the requested table. However, I would like to express our concern regarding the tone of this comment, which we find disrespectful and inappropriate. The implication that the data may not be genuine or that the authors have acted without scientific integrity is both unfounded and inconsistent with the principles and ethics of peer review. 

 As requested, the required data have been provided in the supplementary material in two separate tables: one for glycemia and insulinemia, and another for tumor localization.

comment 2 :Cannot claim arterial phase had highest conspicuity.

response 2: The text has been revised to adopt a less assertive tone.

comment 3:  please add the information above to the methods. Also, how were masses at the border of the transition from the right pancreatic limb to the body and from the left pancreatic limb to the body classified? Were they classified as being within the corpus if part of the insulinoma was in the body or did e.g. >50% of the insulinoma needed to be present int he pancreatic body?

response 3:  In the Materials and Methods, the morphological classification of the lesions has been rewritten more clearly.

comment 4: Reporting lymphadenomegaly is not meaningful without histopathology.

response 4: Considering that six out of 70 patients had cytological confirmation of lymph node involvement in insulinoma, completely excluding the discussion of lymph nodes could make the text less comprehensible, as they would be mentioned only in the cytological results without context. Therefore, in the Discussion, the section has been revised to clarify that in most of cases lymph node involvement remains only a suspicion of metastasis, and that further studies with cytological or histopathological diagnostics are necessary to evaluate the behavior of metastatic lymph nodes in cases of insulinoma.

Reviewer 3 Report

Comments and Suggestions for Authors

I'm glad that in general all the comments were useful in improving the manuscript.

 I have only few suggestions of minor revisions:

-I would recommend that the authors carefully check the manuscript for typos (e. g pag 3, line 126);

-I would invite the authors to use consistent accuracy in detailed information in 2.2 section (e.g. infusion rate for fluidotherapy);

-The claim that radiation dose is considered "non-essential" in veterinary practice is questionable, and I would urge the authors to take this into consideration in future work as well, and in light of what the authors say in the present (line 76, references 13, 14).

 Computed tomography is common in human medicine and are being utilized increasingly by the veterinary profession as scanners become more available and affordable. Although many veterinary specialists are poorly aware of radiation safety issues and it is generally believed that doses to professionals and members of the public from these applications will be very low or negligible and that doses to animals will not be particularly harmful or affect their lifetime risk of developing cancer, the AVMA has emphasized the need to consider the adverse effects of ionizing radiation other than cancer in both humans and small animals and to improve the knowledge of various veterinary specialists regarding the effective patient dose, the risk of ionizing radiation to veterinary patients, and occupational radiation safety issues.

Adding radiation dose details and refining them with new diagnostic methodologies can help foster a cultural shift in the veterinary profession's approach to ionizing radiation imaging toward a holistic "One Health, One Medicine" approach to radiation protection, which could further enhance the quality of veterinary publications.

Author Response

comment 1: I would recommend that the authors carefully check the manuscript for typos (e. g pag 3, line 126);

response 1: Thank you for the correction. The typos have been reviewed.

comment 2:I would invite the authors to use consistent accuracy in detailed information in 2.2 section (e.g. infusion rate for fluidotherapy);

response 2: Thank you for the suggestion; more details have been added.

comment 3:  The claim that radiation dose is considered "non-essential" in veterinary practice is questionable, and I would urge the authors to take this into consideration in future work as well, and in light of what the authors say in the present (line 76, references 13, 14). Computed tomography is common in human medicine and are being utilized increasingly by the veterinary profession as scanners become more available and affordable. Although many veterinary specialists are poorly aware of radiation safety issues and it is generally believed that doses to professionals and members of the public from these applications will be very low or negligible and that doses to animals will not be particularly harmful or affect their lifetime risk of developing cancer, the AVMA has emphasized the need to consider the adverse effects of ionizing radiation other than cancer in both humans and small animals and to improve the knowledge of various veterinary specialists regarding the effective patient dose, the risk of ionizing radiation to veterinary patients, and occupational radiation safety issues. Adding radiation dose details and refining them with new diagnostic methodologies can help foster a cultural shift in the veterinary profession's approach to ionizing radiation imaging toward a holistic "One Health, One Medicine" approach to radiation protection, which could further enhance the quality of veterinary publications.

response 3: In light of the reviewer’s comments, we have also included the radiation dose in the table.

Round 3

Reviewer 2 Report

Comments and Suggestions for Authors

Thank you for your revised submission and for providing the supplementary table. I appreciate your efforts to address the previous comments.

However, I would like to respectfully note that the current supplementary data does not allow reviewers to verify whether all cases included in the study meet the diagnostic criteria for insulinoma. Specifically, not all patients appear to have concurrent glucose concentrations below 63 mg/dL and insulin concentrations above 10 μU/mL, which are standard thresholds for diagnosis.

To ensure clarity and scientific rigor, I kindly request that you revise the supplementary table to include:

  • Concurrent glucose and insulin concentrations for each case
  • Histopathological confirmation status (if applicable)
  • Tumour location

This additional information will help reviewers confirm the inclusion of only confirmed insulinoma cases and strengthen the validity of your study cohort.

Author Response

Comment 1: Thank you for your revised submission and for providing the supplementary table. I appreciate your efforts to address the previous comments. However, I would like to respectfully note that the current supplementary data does not allow reviewers to verify whether all cases included in the study meet the diagnostic criteria for insulinoma. Specifically, not all patients appear to have concurrent glucose concentrations below 63 mg/dL and insulin concentrations above 10 μU/mL, which are standard thresholds for diagnosis. To ensure clarity and scientific rigor, I kindly request that you revise the supplementary table to include:

  • Concurrent glucose and insulin concentrations for each case
  • Histopathological confirmation status (if applicable)
  • Tumour location

This additional information will help reviewers confirm the inclusion of only confirmed insulinoma cases and strengthen the validity of your study cohort.

Comment 1:  

We thank the reviewer for the careful revision and for the further suggestions aimed at strengthening the scientific rigor of the study. The requested variables have now been added to the supplementary table, including concurrent glucose and insulin concentrations, histopathological confirmation status, and tumour location for each case.

Regarding the diagnostic thresholds for insulinoma, we would like to clarify that, as reported in the literature cited as reference 3 (Buishand FO, Current trends in diagnosis, treatment and prognosis of canine insulinoma, Veterinary Sciences, 2022), plasma insulin concentrations may be within the reference range in confirmed cases of insulinoma. This is also reflected in some cases included in the present study, where insulin concentrations were <10 µU/mL despite histological confirmation of insulinoma. Specifically, Buishand states that “…plasma insulin can also be within the reference range,” highlighting that absolute insulin cutoffs alone may not reliably classify all cases.

Moreover, as a tertiary hospital, we frequently receive patients who already carry a prior diagnosis of insulinoma and are referred for further confirmation, staging, or management. Plasma insulin and glucose concentrations may fluctuate, and some dogs may have already initiated pharmacological treatment, as noted in the legend of the supplementary table. These factors can influence the biochemical values measured at presentation.

Additionally, in a recently published epidemiological study (Kraai K., O’Neill D.G., Davison L.J. et al., Incidence and risk factors for insulinoma diagnosed in dogs under primary veterinary care in the UK, Sci Rep 15, 2463 (2025)), insulinoma cases were defined as meeting at least one of the following criteria:

  1. A recorded final diagnosis of insulinoma in the EHRs.

  2. Histopathological confirmation of insulinoma.

  3. Concurrent blood glucose <4.2 mmol/L, plasma insulin >10 µU/mL, and compatible clinical signs of hypoglycemia.

  4. Concurrent hypoglycemia, presence of a pancreatic mass lesion on diagnostic imaging, compatible clinical signs, and absence of systemic inflammatory response syndrome.

In the Authors' opinion, these criteria reflect the heterogeneity of diagnostic pathways encountered in current veterinary practice and support the inclusion of cases that may not meet strict biochemical thresholds but are nonetheless consistent with the accepted diagnostic framework.

We trust that these clarifications, together with the expanded supplementary table, address the reviewer’s concerns and improve the transparency and robustness of the dataset. Please let us know if any further refinements are needed.